# Long-Term Olfactory Memory in African Elephants

**DOI:** 10.3390/ani13040679

**Published:** 2023-02-15

**Authors:** Franziska Hoerner, Arne Lawrenz, Ann-Kathrin Oerke, Dennis W. H. Müller, Idu Azogu-Sepe, Marco Roller, Karsten Damerau, Angelika Preisfeld

**Affiliations:** 1Department of Zoology, University of Wuppertal, 42119 Wuppertal, Germany; 2Zoo Wuppertal, 42117 Wuppertal, Germany; 3Endocrinology Laboratory, German Primate Centre, 37077 Goettingen, Germany; 4Zoological Garden Halle, 06114 Halle (Saale), Germany; 5Serengeti-Park Department of Research, Serengeti-Park Hodenhagen, 29693 Hodenhagen, Germany; 6Zoo Karlsruhe, 76137 Karlsruhe, Germany; 7Department of Ecology, Europa-Universität Flensburg, 24943 Flensburg, Germany

**Keywords:** *Loxodonta africana*, scent memory, long-term memory, human care

## Abstract

**Simple Summary:**

African elephants are known for their long memory; this is also valid for their olfactory sense and their ability to discriminate scents. This feature is highly important for these mammals to maintain their family bonds and to differentiate between familiar and unfamiliar individuals. Thus far, scientific data only testify to an olfactory memory of up to one year for African elephants. This study investigated the long-term olfactory memory of two mother-daughter pairs that were separated for 2 and 12 years, respectively. Results showed that all four elephants were able to recognize their separated relatives just by the scent of feces, thereby giving the empirical implication of olfactory memory in African elephants of up to 12 years.

**Abstract:**

African elephants are capable of discriminating scents up to a single changed molecule and show the largest reported repertoire of olfactory receptor genes. Olfaction plays an important role in family bonding. However, to the best of our knowledge, no empirical data exist on their ability to remember familiar scents long-term. In an ethological experiment, two mother-daughter pairs were presented with feces of absent kin, absent non-kin, and present non-kin. Video recordings showed reactions of elephants recognizing kin after long-term separation but only minor reactions to non-kin. Results give the empirical implication that elephants have an olfactory memory longer than 1 year and up to 12 years and can distinguish between kin and non-kin just by scent. These findings confirm the significance of scent for family bonds in African elephants.

## 1. Introduction

African elephants use a complex olfactory system to discriminate between scents to one changed molecule and have the largest repertoire of olfactory receptor genes reported so far [1,2,3]. This allows them to detect resources and find mating partners in the wild [4,5,6,7,8,9]. It is assumed that the olfactory sense of elephants is also essential for maintaining their family bonds [10], which is an important trait for herds, considering the strenuous environment of the African drylands with its limited resources [11,12]. Olfaction is, next to vision, not only important for the recognition of relatives but also triggers the release of the hormone oxytocin, which stimulates the mother-child bond [13,14]. Elephant bonds are among the strongest in mammals, and especially the relationship between mothers and daughters is the longest, closest, and most intense of all known family bonds [2,11]. The capacity to recognize long-time absent and even mortal remains of relatives is hypothesized to be the result of their complex olfactory abilities [2,15,16]. However, there are no empirical data giving evidence for an olfactory memory in African elephants longer than one year, neither under ex situ nor in situ conditions [3]. It is the ex situ environment that holds the opportunity to investigate elephants’ olfactory abilities under controlled and reliable conditions as it allows for an artificial setting, which is barely possible in wildlife environments [17,18]. 

Species-specific behavior, as an indicator of animal welfare, is a crucial aspect when it comes to zoo-kept elephants and is frequently applied as evidence for animal well-being [19,20]. At the same time, species-specific reactions by zoo elephants to a certain scent, e.g., when smelling familiar scents or finding scents in a new setting, are signs of the animals’ natural development and behavior, similar to their conspecifics in the wild, and are therefore valued as a welfare-indicator [15,19,20]. An additional tool to determine animal welfare is the determination of concentrations of glucocorticoids, often used for assessing physiological stress in elephants, as stress triggers the release of cortisol (glucocorticoids) [21].

In this study, a feces-smelling test was developed as a new tool for the investigation of the olfactory abilities of elephants. The examination group consisted of four zoo-kept female African elephants, two mothers, and their two daughters. The aim was to answer the following research questions: (i) do elephants differentiate between family members and non-kin just by the scent of their feces? (ii) does the scent recognition exceed a separation period longer than the one year reported before [3]? (iii) does the familiar social behavior of elephants under human care, regarding the expression of excitement or indifference, agree or correspond with that of elephants living in situ? (iv) do the tests cause any physiological stress measurable as fecal glucocorticoid metabolites before and after the presentation of the fecal samples? (v) is there a difference between mothers and daughters in reaction to scent recognition?

The test settings described here were performed to predict the reactions of the elephants in two planned re-unifications of the mother-daughter pairs of this project, which were separated for 2 and 12 years, respectively. Since experiences with such transfers are missing, the tests are also of practical use for future elephant associations after transports.

## 2. Materials and Methods

### 2.1. Animals and Designs of the Study

In 2020, the European studbook for African elephants recommended the re-unification of two mother-daughter groups living separately in three German zoos. Information about the elephants is shown in Table 1:

### 2.2. Test Setting

This study was performed as a pre-test for the planned re-unifications of the two mother-daughter pairs, which was later also monitored scientifically [22]. To evaluate elephant reactions before re-unifications and to test their olfactory memory, all four elephants were presented with three different fecal samples beforehand, resulting in three trials conducted with each of the four elephants: (a) an absent kin sample from a separated mother or daughter, (b) a present non-kin sample from an unrelated female, and (c) an absent non-kin sample from a female elephant that all the observed elephants had never met before and were therefore entirely unfamiliar with. The study was designed according to Bates et al. [15]. 

All trials were performed under the same testing conditions. The samples were collected in the morning hours between 7 to 8 a.m. and used within 24 h. The diet of all cows, whose feces were used, was the same. 

The samples were presented separately to the elephants. As all three trials were conducted with the four elephants, there was a total number of 12 trials (*n* = 12). Samples were presented in the following order: (1) present non-kin, (2) absent non-kin, (3) absent kin. The assumingly least interesting sample (present non-kin) was presented first, and the assumingly most interesting sample (absent kin) was presented last to prevent any overshadowing of the reactions. For each elephant, the three trials were conducted on the same day in the same enclosure. For each trial, the fecal sample was placed in a prominent place within the enclosure. Then, gates were opened, and the elephant was granted access to the enclosure where the sample was placed. After each trial, a break of at least two hours was taken before the next trial.

Elephants were alone during the trials. The testing area was cleaned after each trial so that there were no leftovers of the scent of the previous sample. The whole experiment was conducted once for each animal. The test time for each sample presentation was limited to twenty minutes since, after this time, elephants showed no new reaction. 

### 2.3. Data-Collection

All trials were video-recorded with two video cameras of the type Panasonic HTC-TM60, which were placed around the enclosure to cover every area of the enclosures so that the elephants’ reactions were recorded at all times during trials. The test setting can also be seen in the Appendix A (Reaction to olfactory samples a–c by elephant cow PORI).

Elephant reactions to sample presentation were documented by scan sampling with an ethogram derived from Poole and Granli [23,24], which consists of 27 behaviors (see Table 2). These behaviors were aggregated into four higher-level behavioral categories: neutral, excitement, mental processing, and sample examination. 

The time for which elephants showed a certain behavioral category during the sample presentation was measured to analyze the animals’ general reactions to the different samples. It was also observed how many different behaviors of excitement were shown simultaneously during the different trials to investigate the level of excitement the different scents caused in the elephants. Additionally, as an indicator of the rate with which elephants reacted, how many shifts in behavior elephants showed during trials was counted.

Videos were analyzed with a focus on the behaviors of measurement. The ethological data were collected by human observation [17,18,25,26,27]. Extracts from the videos displaying the behavioral reaction of elephant cow PORI can be seen in the Appendix A (Reaction to olfactory samples a–c by elephant cow PORI).

To evaluate if the experiment possibly caused physiological stress in the four study animals, five fecal samples were taken from each elephant. One control sample was collected in the morning before the trials, another sample 24 h later, in the morning after the trials, and then three more samples were obtained in 12 h intervals, with the last being collected in the evening of the second day after the trials. This protocol was used since the main metabolite of cortisol, 11-oxo-etiocholanolone (11-oxo-CM), is only excreted 24 h after a stress event in African elephant feces [28] and compensates for the diurnal variation of cortisol, with higher levels in the morning and decreasing values over the day.

### 2.4. Data Analysis

The data collection resulted in four sets of data: (1) the reaction of the elephants to the different samples, according to the four behavioral categories (Table 2); (2) the number of simultaneously shown behavior by elephants; (3) the number of shifts in behavior shown by elephants; and (4) the changes in the 11-oxo-CM level in the elephants’ feces before and after the experiment. The amount of time elephants showed a certain behavior (dataset (1)) was normalized to a joint maximum of 100% [29,30].

Statistical analysis for all data was carried out with IBM SPSS Statistics 29. An analysis of the graphical distribution for all four datasets determined that datasets (1) and (4) were not normally distributed, and datasets (2) and (3) were normally distributed. Statistical tests were chosen accordingly [31,32]. The significance level for all tests was set at *p* ≤ 0.05 [33,34,35]. 

For dataset (1), the Friedman test for non-normally distributed datasets with more than two connected samples was calculated to detect significant differences between behavioral reactions to fecal samples (a)–(c). If significant differences were detected, a post hoc test with the Bonferroni correction was calculated to determine the significance [36,37]. For differences in the simultaneously shown behavior (dataset (2)) and shifts in behavior (dataset (3)), an ANOVA for normally distributed data was conducted [31]. The changes in 11-oxo-CM level in the elephant feces before and after the experiment (dataset (4)) were analyzed utilizing the Wilcoxon signed-rank test [32]. 

An overview of the statistical tests run for all sets of data and their dependent and independent variables are given in Table 3:

## 3. Results

As shown in Figure 1, the time spent on the presented fecal samples of absent kin was significantly higher in the active response categories of excitement, mental processing, and sample examination. Time spent on the neutral reaction was significantly lower for samples of absent kin. Present and absent non-kin caused fewer reactions, and less time was spent on the active response categories, while neutral behavior was distinctive.

During tests, elephants expressed all behavior of excitement and mental processing when presented with samples from absent kin. However, they did not show particular interest in the scent of absent or present non-kin. Statistical for this analysis can be seen in Table 4.

Examination of the difference in reaction toward the sample of the absent kin between mothers and daughters revealed that mother elephants reacted with a wider variety in their behavioral response as compared to daughter elephants.

As demonstrated in Figure 2, mothers showed up to eleven excitement behaviors simultaneously, while daughters only showed two to three behaviors at the same time (ANOVA: *F*(1,2) = 289.0, *p* = 0.003). Mothers performed 55–64 shifts in behavior, while daughters showed a less intense reaction with just 15–16 shifts (ANOVA: *F*(1,2) = 94.44, *p* = 0.01) (Figure 3).

11-oxoetiocholanolone (11-oxo-CM) was measured as the main metabolite of cortisol in African elephant feces [21]. While all elephants had individual cortisol levels, none reacted with a measurable increase in physical stress after the sample presentations within the following two days. 11-oxo-CM varied before and after the olfactory test for the four elephants from 966.91 to 728.09 ng/g feces (mother elephant BIBI, age 35), 659.89 to 644.9 ng/g feces (daughter elephant PANYA, age 13), 1216.00 to 914.29 ng/g feces (mother elephant PORI, age 39) and 759.71 to 593.26 ng/g feces (daughter elephant TANA, age 19), respectively. The statistical analysis of the glucocorticoid metabolite revealed no significant changes in levels in correlation to the olfactory tests for any of the elephants (Wilcoxon signed-rank test: *z* = 1.342, *p* = 0.180).

## 4. Discussion

Even though this study was performed only on four elephants, data suggest that elephants can recognize the scent of their relatives after up to 12 years of separation. It thereby also demonstrates the intense social bond between elephants [11,12,23]. Even after 12 years of absence, the scent of a relative caused reactions of excitement (see also Appendix A). 

The data also indicate the capacity of zoo-kept elephants to discriminate between kin and non-kin feces and scent, respectively. This is reflected by the time and quantity of reactions and interest expressed toward the sample of their absent kin. The elephants studied here exhibited all behavioral categories associated with agitation and connected the scent to their relative, whereas only minor interest and agitation but major neutral behavior was shown during the non-kin sample presentation. The excitement behavior (Table 2) that was observed during the trials can be assumed as a positive association, as indicated by the expressed rumbling noises and the ear flapping, which are both classified as positive agitation [23,24], and their repeated examination of the sample. Hence, the sample presentation of the absent kin led to a positive reaction of possible emotions, which means that behavioral and bonding concepts by the elephants with their family members are still present ex situ and do not get lost in the zoo environment [19,20]. Missing significances in the differences between absent kin and absent non-kin samples in the categories of sample examination and neutral behavior, despite high differences in expressed behavior, can be traced to the limited sample size of segregated related females [38].

These findings correspond to the situation of elephants in the wild, where the encounter with and differentiation of the scents of herd members or unfamiliar elephants of other herds occur constantly [12,15]. Usually, in situ non-kin scents, as well as common, neutra, and entirely unfamiliar scents, do not cause any major positive reaction or emotional connotation [12,15,39,40,41]. Related elephants rely on olfactory recognition for bonding and herd maintenance [15,41,42]. In this study, the recognition of and reaction to the scent of absent kin exceeded the interest in new or unfamiliar scents, as also described by Bates et al. [15].

Other mammals (e.g., golden hamsters, meadow voles, and humans) are also capable of recognizing kin by scent [43,44,45]. From studies in humans, it was concluded that breastfed newborns recognize their mothers by scent, and mothers, on the other hand, recognize their infant’s scent [45]. Porter [45] states that human relatives have similar scent ‘signatures’ that endorse the recognition of kin. Hence, the findings of the study at hand are supported by comparable information for other mammals. 

In a joint study, Hörner et al. [22] confirmed the positive reaction to the presentation of a relative during the re-unification of mothers and their daughters. However, unrelated elephants living in zoos reacted with tension and agonistic behavior during first encounters as part of unification. This occurs more so than in the wild [12,23], where total spatial avoidance is possible. The means to avoid tension and agonistic behavior and to enhance animal welfare under human care are delimited due to restrictions on the site and research gaps. A prerequisite to further explore these means is the knowledge of elephant stress levels during (re-)unification. The results of this study indicate that the fecal sample presentation did not induce an increase in physiological stress, expressed in the level of glucocorticoids, and can be a potentially useful test in advance of future (re-)unification [21,22].

Interestingly, the data suggest that the mother-offspring bond in elephants is stronger than the offspring-mother bond, as shown in the higher reaction of the mothers to the feces of the absent daughter when compared to the reaction of the daughters to their mothers’ feces. Thus far, no other studies on African elephants have tackled this question; however, research in other mammals with strong family bonds and living in a fission-fusion society, such as chimpanzees, provided similar results [46,47]. In elephants, a possible cause for this reaction is the different relationships mothers and daughters have within elephant herds. Whilst the mothers within a matriarchial group structure seek to protect and keep their family (and thus their daughters) together throughout their entire life, it is common for the daughters to survive their mothers. Thus, losing the mother is a normal (although once-in-a-lifetime) experience. The finding of remains (even scents) of mothers should, therefore, not motivate further reactions. The rediscovery of a lost female offspring, however, may trigger searching behavior, even resulting in stronger behavioral reactions when smelling their scent. Another possible reason for the discrepancy found between mothers and daughters could be the nature of the mother-child relationship in general. Elephant mothers invest their whole life into their offspring, while daughters do not invest in their mothers [48]. Furthermore, both mother elephants of this study have experienced the death of offspring before and, therefore, might react more strongly than the daughters. Additionally, the reactions of mothers could be stronger because they were in an environment with non-family members, whereas the daughters had their own offspring with them at the time of the experiments. Hence, being confronted with the scent of a relative after housing in an environment without any related individuals might trigger more pronounced reactions.

The finding of the mothers reacting stronger than the daughters neglects the findings of other studies that in many mammals, the olfactory sense and also the ability to discriminate scents shows impairment with growing age [49,50,51]. 

Since no increase in 11-oxo-CM was detected in any of the four elephants, it can be assumed that neither mothers nor daughters experienced measurable stress during the smelling test despite all observed reactions [21]. This also suggests that experiments similar to this can be performed without affecting the welfare of elephants and are a safe method to be applied in future elephant transfers.

Even though there is evidence for actual short- and long-term olfactory memory in mammals [52], odor memory was considered as ‘primitive’ for a long time [53]. However, studies have already testified to the learning ability in short-term memory for odors in mammals [54], which was also testified for elephants [41]. When it comes to long-term olfactory memory, data for mammals are limited. Noack et al. [55] mention a long-term olfactory memory in mice; however, they only refer to an olfactory memory of up to 7 days, which is relatively short considering the life expectancy of mice, which is approximately two years. Hence, considering the different life expectancies of mice and elephants, these findings of long-term memory in mice by Noack et al. [54] equals the long-term memory of one year found in elephants before [3,41]. Szenczi et al. [56] at least testify to an olfactory memory for scents, in general, of up to one year in domestic cats. Under this perspective, the findings in the present study can be considered relevant in the broader term of research on long-term olfactory memory in mammals. 

## 5. Conclusions

This report provides empirical evidence for long-term olfactory memory of up to 12 years in African elephants, which is distinguishably longer than the long-term olfactory memory reported for other mammals [55,56]. The study indicates that the reaction to scents of relatives is positive and therefore attests to species-specific behavior in zoo-socialized elephants [23,24]. The long-term olfactory memory and the positive reaction to the relatives’ scent are further confirmations of the close family bonds in African elephants, especially from mother to daughter. The study also promotes a new testing tool for future transfers, which can be used as a method to familiarize elephants with scents before unification, to predict reactions during re-unifications, as described in Hörner et al. [22], and to secure elephant safety and welfare. For future studies, the assessment of animal welfare following re-unifications would help in providing more evidence for the usefulness of the fecal-smelling test.

## Figures and Tables

**Figure 1 animals-13-00679-f001:**
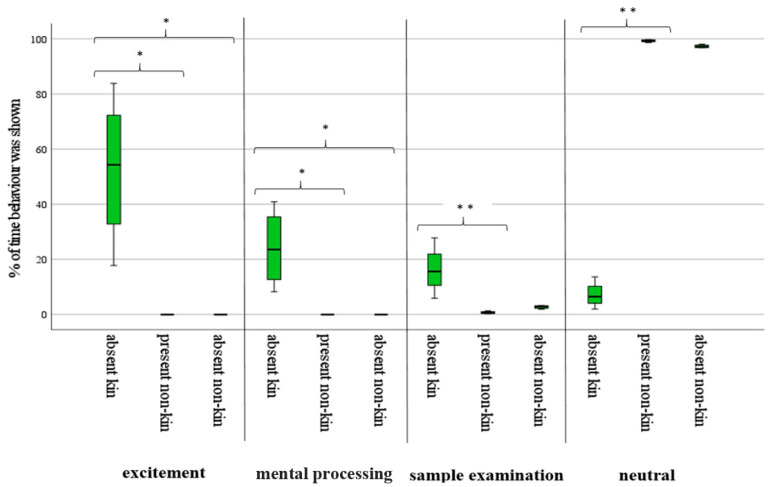
Reaction (in percentage of time) shown after confrontation with fecal samples (* *p* < 0.05; ** *p* < 0.01).

**Figure 2 animals-13-00679-f002:**
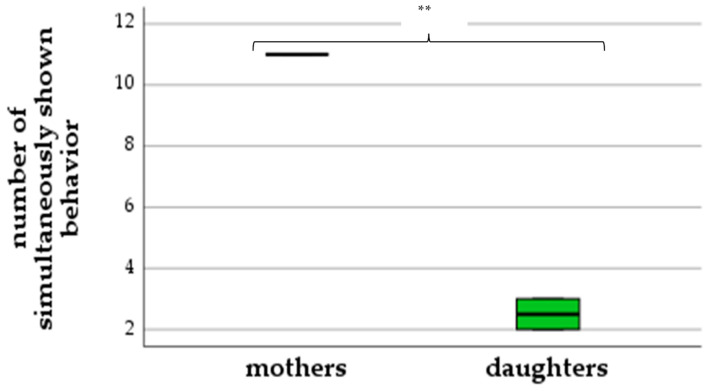
Number of simultaneously shown behaviors (differences between mothers and daughters) (** *p* < 0.01).

**Figure 3 animals-13-00679-f003:**
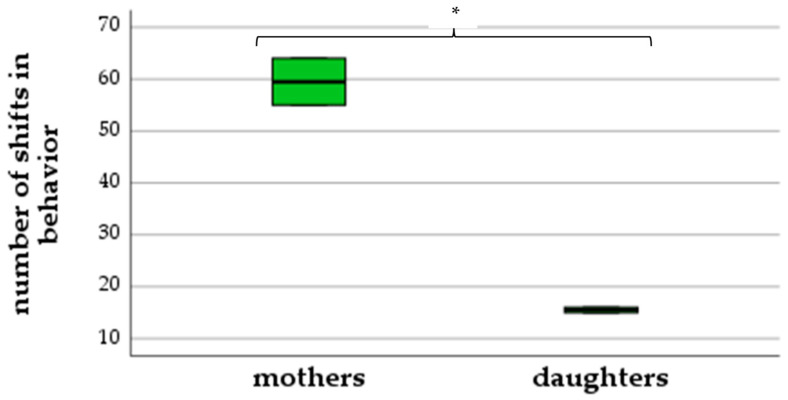
Number of shifts in behavior (differences between mothers and daughters), (* *p* < 0.05).

**Table 1 animals-13-00679-t001:** Record of elephants of the study.

	Sex	Place of Birth	Date of Birth	Transferred From and To	Relation	Living with Other Relatives	Years Living Separated
BIBI	F	Zimbabwe	1985	-	Mother	No	2
PANYA	F	Tierpark Berlin	22 August 2007	Bergzoo Halle to Serengeti-Park Hodenhagen	Daughter	Yes ^1^	
PORI	F	Zimbabwe	1981	Tierpark Berlin to Bergzoo Halle	Mother	No	12
TANA	F	Tierpark Berlin	4 May 2001	-	Daughter	Yes ^2^	

^1^ Living with a son. ^2^ Living with two daughters.

**Table 2 animals-13-00679-t002:** Behaviors during sample presentations and their categories (extracted from Poole and Granli [23,24]).

Category	Behavior
Neutral	Walking around the enclosure
	Trunk locomotion
	Eating
	Body care
	Comfort behavior
	Weaving
Excitement	Folding, lifting, spreading, flapping ears
	Raising trunk
	Shaking trunk
	Raising head
	Shaking head
	Raising tail
	Shaking tail
	Pacing
	Pacing backward
	Rumbling
	Defecating and urinating
	Glandular secretion
	Throwing feces
	Intense weaving
Mental processing	Freezing
	Listening
	Smelling air
Sample examination	Sniffing on sample
	Examining sample with trunk and/or feet
	Squashing sample
	Throwing sample

**Table 3 animals-13-00679-t003:** Overview of statistical analysis.

Set of Data	Distribution	Statistical Test	Dependent Variable	Independent Variable
(1) Behavioral reaction to samples	Not normal	Friedman test + post hoc test	Time spent reacting to the presented fecal samples	Different fecal samples
(2) Simultaneously shown behavior	Normal	ANOVA	Number of shifts in behavior	Mother/daughter
(3) Shifts in behavior	Normally	ANOVA	Number of simultaneously shown behaviors	Mother/daughter
(4) Changes in 11-oxo-CM level	Not normally	Wilcoxon signed-rank Test	Level of 11-oxo-CM	Time of measurement

**Table 4 animals-13-00679-t004:** Corresponding statistics after Friedman tests and post hoc tests.

		Excitement	Mental Processing	Sample Examination	Neutral
N	4				
df	2				
Asymp. Sig.	0.018				
Sig.	a: abs. kin/pres. n-kin	0.034	0.034	0.005	0.005
	b: abs. kin/abs. n-kin	0.034	0.034	0.157	0.157
	c: pres. n-kin/abs. n-kin	1.0	1.0	0.157	0.157

## Data Availability

Data available on request due to restrictions eg privacy or ethical. The data presented in this study are available on request from the corresponding author. The data are not publicly available due to ongoing data curation.

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
