# Peer review of "Long-Term Olfactory Memory in African Elephants"

_animals, 2023, doi:10.3390/ani13040679_

Round 1
Reviewer 1 Report
I found this paper somewhat difficult to review, as there is definitely data of interest (although this is a very small amount of data, i.e. 12 data points) that does have applicability, but the framing of this research by the authors distracts from the useful information they have actually generated.
Let me start by saying the overall concept is good, the execution of the study is good, the analysis has been done well, and the non-hyped results useful. These results may improve the manner in which elephants are moved, especially related elephants, between zoos. This is important.
It would be appropriate to include a figure of the results comparing the mother vs daughter pairs in a manner somewhat similar to that presented in Fig 1, with the individual animals. Given there are only 4 individuals, the more the data is presented the easier it is for the reader to extract the truly relevant information. This is the only concern I have regarding the actual science of the paper.
The remainder of my comments focus on the presentation and interpretation of the ideas and data. The authors may disagree with me on several points, but I feel I must raise these points and will specifically state what I mean.
I'll start from the beginning and move through the paper.
"African elephants are known for their impressive memory". Really? In what sense is their memory "impressive"? Impressive, as a word, is hyperbole, and hyperbole are statements or claims not meant to be taken literally. Hyperbole has not place in science as scientific writing is meant to be taken literally. As far as I am aware, the potential capacity and clarity of elephant memory has not been explored in detail, and from what we know, their capacity and clarity for episodic and autobiographical memory likely does not exceed what one would expect for a long-lived terrestrial mammal (see 10.1007/s00429-013-0587-6). Is this really "impressive"?
"Family bonds". This term appears repeatedly throughout the manuscript. The evidence for odours playing a role in family bonding, or rather, more specifically, recognition of individuals and the animal's recollection of the significance that such an individual has to them, is a common feature across mammals and probably other vertebrates. Olfaction is a sense involved in a huge range of behaviours, although recognition of individuals is one important aspect. Family bonding is a far more complex behaviour than recognition of individuals. The authors would be served well to explain this properly at first, and then talk about family bonding in this context. It should also be noted that in line 3 of the introduction the authors say that this is "assumed" to be "essential" for family bonds. I hope the disconnect I am trying to highlight is clear. Elephants appear to bond ("bonding" being reflected in the specific release of oxytocin) using the tactile sense, so olfaction and vision might recognize and individual, but tactile stimulation (through touching with the trunk, etc) is likely what is stimulating bonding. The authors need to understand the concepts they using...however, if they can cite papers showing that odours release chemicals that form social bonds, even in other mammals, I will withdraw what I state above.
"First empirical proof". The use of the descriptor "first" is a growing trend in the "@#amazingscience" world. Every scientific paper, unless a review or repeat of a previous study, should be a "first". This is what deriving original data is about. It is implicit that the original data is first, and doesn't need to be stated, repeatedly, in a scientific paper. Please remove all references to first, the publication of the manuscript will give it the priority the authors appear determined to seek.
"Discriminating scents on a molecular level". This is shows a complete lack of understanding of how the sense of olfaction actually works. Olfaction, in all species, works through the detection of water-borne molecules that stimulate receptors within olfactory nerve terminals. So to state that "Elephants are capable of discriminating scents on a molecular level..." is stating how we all know the sense of olfaction works. My reading of this is that the authors have not actually investigated in any detail the sense they are testing, and this is disconcerting. Nor have they examined previous studies related to olfaction in the elephant and how this might be processed by the brain. These studies, 10.1007/s00429-01100316-y and 10.1007/s00429-017-155-3, especially the latter is important in terms of understanding the role of olfaction in recognition of related and non-related indivuduals.
Line 2 of introduction, the phrase "to precisely detect resources". "precise" is a very specific word with a very specific meaning. Do elephants detect resources at the molecular scale? Or are they rather getting a rough idea of where a resource is and head in that direction? Either be precise, or remove this word.
In the discussion: "Related elephants rely on olfactory recognition for bonding and herd maintenance". The papers cited to support this statement by Archie et al (2006) and Moss (2001) did not investigate olfaction. I searched both articles for the terms "olfaction" "smell" and "odour/odor" and found no occurrences. Thus, the statement made by the authors is not supported by the papers cited. This leads me to wonder whether the authors actually read the papers they cited?
"a great monitoring" the word great is absolutely subjective. Why not use the simpler and much less loaded wording "potentially useful".
Change "gives clear" to "provides".
Author Response
Response to Reviewer 1 Comments
Point 1: It would be appropriate to include a figure of the results comparing the mother vs daughter pairs in a manner somewhat similar to that presented in Fig 1, with the individual animals. Given there are only 4 individuals, the more the data is presented the easier it is for the reader to extract the truly relevant information. This is the only concern I have regarding the actual science of the paper.
Response 1: Figures of the data for differences between mothers and daughters were added. We chose boxplots, as here data for both mothers and both daughters can be presented in one. See page 6.
Point 2: "African elephants are known for their impressive memory". Really? In what sense is their memory "impressive"? Impressive, as a word, is hyperbole, and hyperbole are statements or claims not meant to be taken literally. Hyperbole has not place in science as scientific writing is meant to be taken literally. As far as I am aware, the potential capacity and clarity of elephant memory has not been explored in detail, and from what we know, their capacity and clarity for episodic and autobiographical memory likely does not exceed what one would expect for a long-lived terrestrial mammal (see 10.1007/s00429-013-0587-6) . Is this really "impressive"?
Response 2: The word was erased.
Point 3: "Family bonds". This term appears repeatedly throughout the manuscript. The evidence for odours playing a role in family bonding, or rather, more specifically, recognition of individuals and the animal's recollection of the significance that such an individual has to them, is a common feature across mammals and probably other vertebrates. Olfaction is a sense involved in a huge range of behaviours, although recognition of individuals is one important aspect. Family bonding is a far more complex behaviour than recognition of individuals. The authors would be served well to explain this properly at first, and then talk about family bonding in this context. It should also be noted that in line 3 of the introduction the authors say that this is "assumed" to be "essential" for family bonds. I hope the disconnect I am trying to highlight is clear. Elephants appear to bond ("bonding" being reflected in the specific release of oxytocin) using the tactile sense, so olfaction and vision might recognize and individual, but tactile stimulation (through touching with the trunk, etc) is likely what is stimulating bonding. The authors need to understand the concepts they using...however, if they can cite papers showing that odours release chemicals that form social bonds, even in other mammals, I will withdraw what I state above.
Response 3: Elephants use their tactile and visual senses to form social bonds, but also olfaction. Some papers state that odors release oxytocin, which results in an “increase in the granule cell layer of the olfactory bulb” (Kendrick, 2000), here talking of sheep. Other papers state that mammals depend on “olfactory information for mother–infant recognition” (Curley & Keverne, 2005) and stimulate the mother-child bond thereby. We explained and clarified this further in the manuscript and hope that it is understandable now. See lines 41-43: “Olfaction is, next to vision, not only important for the recognition of relatives but also triggers the release of the hormone oxytocin, which stimulates the mother-child bond [13-14].”
Point 4: "First empirical proof". The use of the descriptor "first" is a growing trend in the "@#amazingscience" world. Every scientific paper, unless a review or repeat of a previous study, should be a "first". This is what deriving original data is about. It is implicit that the original data is first, and doesn't need to be stated, repeatedly, in a scientific paper. Please remove all references to first, the publication of the manuscript will give it the priority the authors appear determined to seek.
Response 4: “first” was removed.
Point 5: "Discriminating scents on a molecular level". This is shows a complete lack of understanding of how the sense of olfaction actually works. Olfaction, in all species, works through the detection of water-borne molecules that stimulate receptors within olfactory nerve terminals. So to state that "Elephants are capable of discriminating scents on a molecular level..." is stating how we all know the sense of olfaction works. My reading of this is that the authors have not actually investigated in any detail the sense they are testing, and this is disconcerting. Nor have they examined previous studies related to olfaction in the elephant and how this might be processed by the brain. These studies, 10.1007/s00429-01100316-y and 10.1007/s00429-017-155-3, especially the latter is important in terms of understanding the role of olfaction in recognition of related and non-related indivuduals.
Response 5: Our phrasing here is, indeed, misleading. What we wanted to state is, that elephants are capable to discriminate scents up to single changed molecules (Rizvanovic, 2013). We rephrased this and hope that it is clearer now. See lines 24 and 36: “African elephants are capable of discriminating scents up to a single changed molecule”, “African elephants use a complex olfactory system to discriminate between scents to one changed molecule”
Point 6: Line 2 of introduction, the phrase "to precisely detect resources". "precise" is a very specific word with a very specific meaning. Do elephants detect resources at the molecular scale? Or are they rather getting a rough idea of where a resource is and head in that direction? Either be precise, or remove this word.
Response 6: “precise” was removed.
Point 7: In the discussion: "Related elephants rely on olfactory recognition for bonding and herd maintenance". The papers cited to support this statement by Archie et al (2006) and Moss (2001) did not investigate olfaction. I searched both articles for the terms "olfaction" "smell" and "odour/odor" and found no occurrences. Thus, the statement made by the authors is not supported by the papers cited. This leads me to wonder whether the authors actually read the papers they cited?
Response 7: Thanks a lot for pointing this out. This was cited wrong. We revised it.
Point 8: "a great monitoring" the word great is absolutely subjective. Why not use the simpler and much less loaded wording "potentially useful".
Response 8: “great monitoring” was exchanged for “potentially useful”.
Point 9: Change "gives clear" to "provides".
Response 9: Revised.

Reviewer 2 Report
I would like to thank the authors for the opportunity to review this interesting study. This research is important to our understanding of both elephant cognition and scent-signalling in mammals so it is with great pleasure that I read the paper. The experiment itself is sound but I would like more details throughout on how it was undertaken.
I like the introduction’s information on elephants. However, it would benefit from a more thorough review of existing literature on scent recognition and use in social relationships.
Methods – while this is mostly clear, I am a little confused about the presentation of the experiments in line 80 on. There were 12 tests, not 12 experiments, as you had 4 repeats for the 3 experiments I think? A table of what was done when and with whom would probably help here. The inclusion of the comparison of mothers and daughters in the results is very interesting but needs mentioning in the methods so the reader understands what was done.
Some general details on how the experiment was done would help here. How long were the faeces stored for and how were they presented? If the elephants were given 20 minutes to interact, what were the faeces presented in and how long were gaps between trials?
Results – these are fine, though I strongly feel it would help to have a video of the elephants’ behaviour on finding the faeces to help show the difference between investigations.
Discussion – This is generally good but there are two places where it seems to go beyond the data and to overstate a supposition:
Line 183 – I really don’t think you can make this statement like this. The elephants were excited by the discovery of the familiar scent, yes, but it’s not necessarily happiness that they are exhibiting. It is also possible that they are excited but then distressed to not find the individuals present otherwise. This would not necessarily be enough to be reflected in faecal cortisol. If you can link the behaviours they displayed to behaviours linked to other positive experiences, I am happy to revise my concern here.
Again, at line 209 – Or the daughters were less effective at learning their mothers’ scents? Again, this seems to go a step beyond the data in its assertion. Considering the rate of mortality for calves, it seems likely that many elephant mothers would lose at one of their offspring across their lifetime, so it seems odd to argue that the mothers will have a stronger reaction to finding lost offspring than vice versa. The sample size here is also small enough to suggest there may be other things going on.
I don’t think you are necessarily wrong on these points, but I would word these statements more tentatively considering the small sample size and the other potential explanations for the behaviour.
The discussion could also include a little on possible chemical analysis to determine what encodes individuality in the faeces and whether this varies by bloodline, or a discussion of the possibilities here.
Finally, I can’t find an ethics statement. If ethics were waived, please explain how. It would be unusual for zoos to allow any study of their elephants without one, even one as non-invasive as this.
Overall this paper is well presented and with interesting data, it just needs a few points clarified. The experiment itself is fascinating, and the results are valuable to the field. So I'd like to congratulate the authors on producing such an interesting work!
Author Response
Response to Reviewer 2 Comments
Point 1: I like the introduction’s information on elephants. However, it would benefit from a more thorough review of existing literature on scent recognition and use in social relationships.
Response 1: We added information on that. See line 41ff.
Point 2: Methods – while this is mostly clear, I am a little confused about the presentation of the experiments in line 80 on. There were 12 tests, not 12 experiments, as you had 4 repeats for the 3 experiments I think? A table of what was done when and with whom would probably help here. The inclusion of the comparison of mothers and daughters in the results is very interesting but needs mentioning in the methods so the reader understands what was done.
Response 2: We edited the methods section, see 2.2 Test setting (line 85) and 2.3 Data-Collection (line 117). Right now, it is not in form of a table. However, if you still feel like it would be more comprehensible with a table, we will happily provide one.
Point 3: Some general details on how the experiment was done would help here. How long were the faeces stored for and how were they presented? If the elephants were given 20 minutes to interact, what were the faeces presented in and how long were gaps between trials?
Response 3: We added this information, see 2.2 Test setting (line 85).
Point 4: Results – these are fine, though I strongly feel it would help to have a video of the elephants’ behaviour on finding the faeces to help show the difference between investigations.
Response 4: We will happily provide exemplary videos of the trials. I will talk to the editorial office about this.
Point 5: Line 183 – I really don’t think you can make this statement like this. The elephants were excited by the discovery of the familiar scent, yes, but it’s not necessarily happiness that they are exhibiting. It is also possible that they are excited but then distressed to not find the individuals present otherwise. This would not necessarily be enough to be reflected in faecal cortisol. If you can link the behaviours they displayed to behaviours linked to other positive experiences, I am happy to revise my concern here.
Response 5: We edited this section. We would argue that the excitement, displayed by the cows, had a positive connotation, as they expressed signs of excitement which are often referred to as ‘positive’. We elaborated on this in the Material and Methods section and the discussion. We hope that our point becomes more clear. If you still have doubts, we will adjust the manuscript accordingly. See line 208 ff.
Point 6: Again, at line 209 – Or the daughters were less effective at learning their mothers’ scents? Again, this seems to go a step beyond the data in its assertion. Considering the rate of mortality for calves, it seems likely that many elephant mothers would lose at one of their offspring across their lifetime, so it seems odd to argue that the mothers will have a stronger reaction to finding lost offspring than vice versa. The sample size here is also small enough to suggest there may be other things going on.
I don’t think you are necessarily wrong on these points, but I would word these statements more tentatively considering the small sample size and the other potential explanations for the behaviour.
Response 6: We changed this section and gave further explanations for the different reactions between mothers and daughters. See line 248 ff.
Point 7: The discussion could also include a little on possible chemical analysis to determine what encodes individuality in the faeces and whether this varies by bloodline, or a discussion of the possibilities here.
Response 7: We edited the discussion accordingly and also added some further literature. See line 265 ff.
Point 8: Finally, I can’t find an ethics statement. If ethics were waived, please explain how. It would be unusual for zoos to allow any study of their elephants without one, even one as non-invasive as this.
Response 8: An ethics statement was added (see line 316) and also send to animals.

Reviewer 3 Report
This study investigated the responses of elephants to the scent of kin after a lengthy period of separation. The authors presented 4 zoo elephants with dung from absent kin, absent non-kin, and present kin. They found that subjects reacted more strongly to the dung of absent and previously familiar kin than to the dung of present and familiar non-kin or absent and unfamiliar non-kin, indicating that at least these elephants can recognize kin from purely olfactory cues after a period of separation of 2-12 years. They discuss implications for using similar tests to assess captive elephant welfare and reunite zoo elephants after extensive separation.
Overall, I think that this study represents a valuable contribution to our understanding of elephant behavior. While the sample size was extremely limited due to the constraints of working with zoo elephants, this is the first study to my knowledge to provide evidence for such long-term olfactory recognition in elephants, and the study also provides proof of concept for a novel behavioral assay with implications for captive elephant welfare; namely, using dung presentations to assess the feasibility of reuniting former kin/social companions who have been separated for an extensive period of time.
My primary critique of the manuscript is the lack of clarity in the description of the statistical analyses. I find the description of the statistical analyses very difficult to follow, and it is not clear to me exactly what models were run and what the dependent and independent variables were for each model. This needs to be clarified before I can assess the appropriateness of all the methods. In addition to re-writing the text of the methods and results to make it easier to follow, I suggest also including a table with all the models and each of their dependent and independent variables to help make clear exactly what analyses were run.
A more minor comment is that some of the word choice in the manuscript is not ideal. For example, referring to the evidence in this manuscript as “proof” of anything is not appropriate, as proof implies a much higher standard of evidence than what is generally possible to achieve in most empirical studies, let alone one with such a small sample size. Instead of saying that this study “proves” something, it would be better to say that the study provides evidence for, suggests, or implies. When you have very strong evidence for something it may be appropriate to use stronger words like “indicate” or “demonstrate”, but the word “proof” should generally be reserved for mathematics. Other word choices in the manuscript are not scientifically problematic but sound awkward in English; for example, “behavioral item” (line 99), “data testify” (line 173), “interlinked the scent” (line 181), etc. It would be better to say “behavior”, “our data suggest”, and “apparently connected the scent”.
Detailed comments:
· Lines 88-89: To me, the word “experiment” implies a series of trials that were conducted to answer a single research question, while the word “trial” refers to each individual replicate of the experiment. I think it would be better to refer to each presentation of a stimulus to an elephant as a trial and refer to the whole study as a single experiment.
· Line 89: Is there a reason why you presented the samples in a random order instead of varying the order systematically between elephants to ensure balance in the order of presentation? Especially with such a small sample size, it seems to me that randomizing the order could result severe imbalance in the order of presentation. Can you provide a table with the order of the trials for each elephant?
· Line 99: Don’t say that the 27 behavioral items were “further divided” into 4 behavioral categories. “Further divided” implies that the 27 behaviors were broken down into finer categories. Instead, it would be better to say that the 27 behaviors were aggregated into 4 higher-level behavioral categories.
· Should make it more clear in the Methods what the response variables and predictor variables were for each model
· Lines 123-124: It isn’t clear what you mean by “all data sets”. What specifically were each of these datasets, and for each dataset, what was the variable that you assessed for normality?
· Line 133: the fecal glucocorticoid concentration before and after the trial is paired data. Paired data that do not meet the assumptions for a paired t-test should be analyzed with a Wilcoxon signed rank test, not a Mann-Whitney U test.
· Lines 135-140: Not necessary to summarize methods at beginning of results, since methods are presented before results
· Lines 156-161: What do you mean by “more vigorously”? And what exactly were the response variables and factors for the ANOVAs that are cited in this paragraph? It isn’t clear to me from the way the paper is written what analyses you did.
· Line 184: how do you know that the behaviors you observed were indicative of positive emotion? I don’t think you have sufficient evidence to determine if the emotions were positive or negative
Author Response
Response to Reviewer 3 Comments
Point 1: My primary critique of the manuscript is the lack of clarity in the description of the statistical analyses. I find the description of the statistical analyses very difficult to follow, and it is not clear to me exactly what models were run and what the dependent and independent variables were for each model. This needs to be clarified before I can assess the appropriateness of all the methods. In addition to re-writing the text of the methods and results to make it easier to follow, I suggest also including a table with all the models and each of their dependent and independent variables to help make clear exactly what analyses were run.
Response 1: The Material & Methods section was rewritten and edited accordingly to clarify the statistical tests that were used. Additionally, a table for all models was provided with details on sets of data.
Point 2: A more minor comment is that some of the word choice in the manuscript is not ideal. For example, referring to the evidence in this manuscript as “proof” of anything is not appropriate, as proof implies a much higher standard of evidence than what is generally possible to achieve in most empirical studies, let alone one with such a small sample size. Instead of saying that this study “proves” something, it would be better to say that the study provides evidence for, suggests, or implies. When you have very strong evidence for something it may be appropriate to use stronger words like “indicate” or “demonstrate”, but the word “proof” should generally be reserved for mathematics. Other word choices in the manuscript are not scientifically problematic but sound awkward in English; for example, “behavioral item” (line 99), “data testify” (line 173), “interlinked the scent” (line 181), etc. It would be better to say “behavior”, “our data suggest”, and “apparently connected the scent”.
Response 2: Phrases were changed as suggested.
Point 3: Lines 88-89: To me, the word “experiment” implies a series of trials that were conducted to answer a single research question, while the word “trial” refers to each individual replicate of the experiment. I think it would be better to refer to each presentation of a stimulus to an elephant as a trial and refer to the whole study as a single experiment.
Response 3: Changed as suggested.
Point 4: Line 89: Is there a reason why you presented the samples in a random order instead of varying the order systematically between elephants to ensure balance in the order of presentation? Especially with such a small sample size, it seems to me that randomizing the order could result severe imbalance in the order of presentation. Can you provide a table with the order of the trials for each elephant?
Response 4: This was phrased misleadingly. The samples were presented to all elephants in the same order: (1) present non-kin, (2) absent non-kin, (3) absent kin. This order however was chosen at random. However, further considerating showed me that this is not quite true. We went from assumingly least interesting to assumingly most interesting. This order was chosen, as we assumed that a rise in the excitement by the sample of the absent kin might have an impact on overall behavior afterwards. We clarified this in the manuscript. See line 99: “Samples were presented in the following order: (1) present non-kin, (2) absent non-kin, (3) absent kin. The assumingly least interesting sample (present non-kin) was presented first and the assumingly most interesting sample (absent kin) last, to prevent any overshadowing of the reactions.”
Point 5: Line 99: Don’t say that the 27 behavioral items were “further divided” into 4 behavioral categories. “Further divided” implies that the 27 behaviors were broken down into finer categories. Instead, it would be better to say that the 27 behaviors were aggregated into 4 higher-level behavioral categories.
Response 5: Changed as suggested.
Point 6: Should make it more clear in the Methods what the response variables and predictor variables were for each model
Response 6: A table was added that provides information on response variables and predictor variables for each test. See line 164.
Point 7: Lines 123-124: It isn’t clear what you mean by “all data sets”. What specifically were each of these datasets, and for each dataset, what was the variable that you assessed for normality?
Response 7: Information was added. See line 147: “The amount of time elephants showed a certain behavior (data set (1)), was normalized to a joint maximum of 100 % [29-30].”
Point 8: Line 133: the fecal glucocorticoid concentration before and after the trial is paired data. Paired data that do not meet the assumptions for a paired t-test should be analyzed with a Wilcoxon signed rank test, not a Mann-Whitney U test.
Response 8: The Mann-Whitney U test was replaced by Wilcoxen test. Thanks a lot for pointing this out!
Point 9: Lines 135-140: Not necessary to summarize methods at beginning of results, since methods are presented before results
Response 9: Revised as suggested.
Point 10: Lines 156-161: What do you mean by “more vigorously”? And what exactly were the response variables and factors for the ANOVAs that are cited in this paragraph? It isn’t clear to me from the way the paper is written what analyses you did.
Response 10: “more vigorously” means with a bigger variety in their behavioral response. We clarified that. See line 178: “Examination of the difference in reaction toward the sample of the absent kin between mothers and daughters revealed that mother elephants reacted with a bigger variety in their behavioral response as compared to daughter elephants.”. Information on variables for different tests was added. See Table 3.
Point 11: Line 184: how do you know that the behaviors you observed were indicative of positive emotion? I don’t think you have sufficient evidence to determine if the emotions were positive or negative
Response 11: We can indeed assume that the reaction had a positive connotation, as the elephants did not express any signs of fear (ears folded back to the head, tail and head held low, shoulders held high, trumpeting) or anger (head and ears high, throwing and stirring sand, roaring) and showed signs of positive agitation (flapping ears, rumbling). However, this was not clarified in the manuscript. We edited the manuscript for a better understanding accordingly. See line 208 ff.

Reviewer 4 Report
The author presents an interesting study on the olfactory long-term memory in African elephants. The bright side of the manuscript is that it provides some useful practical details on the related topic. Although the number of samples is low, the study still provides informative content on kin recognition in African elephants. However, some missing points in the manuscript (mentioned below). Therefore, I would like to make some suggestions to improve the quality of the paper as below:
In the “2.2. Data-Collection” section materials that were used should be explained. What kind of video recorders were used and how the experimental cage/zone was designed? In other words, how was the experimental area environment? Brief information for the experimental set-up and environment would better fit to this section.
The “2.3. Data Analysis” section should be written in more detail. I mean, authors should give brief information for the fallowing sentences;
“Sets of data were classified numerically [33-34]” please explain with 1-2 sentences.
“ANOVA for normally distributed data was conducted [36]”. Which data were normally distributed? and for ANOVA, what was the dependent and independent variables? Statistical analyses should be explained in detail.
The Discussion section should be enriched with a more theoretical interpretation and relate the present results with additional concepts. For instance, the study results can be discussed with different mammalian species’ kin recognition capabilities in the broader context.
Author Response
Response to Reviewer 4 Comments
Point 1: In the “2.2. Data-Collection” section materials that were used should be explained. What kind of video recorders were used and how the experimental cage/zone was designed? In other words, how was the experimental area environment? Brief information for the experimental set-up and environment would better fit to this section.
Response 1: Information has been added. See line 112 ff.
Point 2: “Sets of data were classified numerically [33-34]” please explain with 1-2 sentences.
Response 2: Information has been changed as follows: “The amount of time elephants showed a certain behavior (data set (1)), was normalized to a joint maximum of 100 % is presented in percentage of time [29-30].” (line 147).
Point 3: “ANOVA for normally distributed data was conducted [36]”. Which data were normally distributed? and for ANOVA, what was the dependent and independent variables? Statistical analyses should be explained in detail.
Response 3: Information on which tests were run on which set of data has been added. Also, a table with detailed information on the statistical testing was provided (see Table 3).
Point 4: The Discussion section should be enriched with a more theoretical interpretation and relate the present results with additional concepts. For instance, the study results can be discussed with different mammalian species’ kin recognition capabilities in the broader context.
Response 4: We elaborated the discussion section further and also included other studies on mammals' olfactory memory and kin recognition, as suggested (see line 197). Thank you very much, the discussion section is much improved.

Round 2
Reviewer 1 Report
The authors adequately addressed the concerns raised.
Reviewer 4 Report
The authors corrected and improved the the manuscript.